# The Effect of Hydroxyl on the Superhydrophobicity of Dodecyl Methacrylate (LMA) Coated Fabrics through Simple Dipping-Plasma Crosslinked Method

**Liyun Xu [1], Yu Zhang [2], Ying Guo [2,3], Ruiyun Zhang [4,5], Jianjun Shi [2,3], Yue Shen [1],\* and Jianyong Yu [5]**

1   School of Textile and Clothing, Nantong University, Nantong 226019, China; gangsheng1989@126.com
2   Department of Applied Physics, College of Science, Donghua University, Shanghai 201620, China; 2161508@mail.dhu.edu.cn (Y.Z.); guoying@dhu.edu.cn (Y.G.); jshi@dhu.edu.cn (J.S.)
3   Member of Magnetic Confinement Fusion Research Center, Ministry of Education, Donghua University, Shanghai 201620, China
4   Key Laboratory of Textile Science and Technology, Ministry of Education, College of Textiles, Donghua University, Shanghai 201620, China; ryzhang@dhu.edu.cn
5   Innovation Center for Textile Science and Technology, Donghua University, Shanghai 201620, China; yujy@dhu.edu.cn
\*   Correspondence: shen.y@ntu.edu.cn

**Abstract:** In order to obtain stable superhydrophobicity, suitable hydrophobic treatment agents should be selected according to different material properties. In this paper, cotton and poly(ethylene terephthalate) (PET) fabrics were respectively coated with dodecyl methacrylate (LMA) via argon combined capacitively coupled plasma (CCP), and the surface hydrophobicity and durability of the treated cotton and polyester fabrics are also discussed. An interesting phenomenon happened, whereby the LMA-coated cotton fabric (Cotton-*g*-LMA) had better water repelling and mechanical durability properties than LMA-coated PET fabric (PET-*g*-LMA), and LMA-coated hydroxyl-grafted PET fabrics (PET fabrics were successively coated with polyethylene glycol (PEG) and LMA, PET-*g*-PEG & LMA) had a similar performance to cotton fabrics. The water contact angles of Cotton-*g*-LMA, PET-*g*-LMA and PET-*g*-PEG & LMA were 156°, 153° and 155°, respectively, and after 45 washing cycles or 1000 rubbing cycles, the corresponding water contact angles decreased to 145°, 88°, 134° and 146°, 127° and 143°, respectively. Additionally, thermoplastic polyurethane (TPU) and polyamides-6 (PA6) fabrics all exhibited the same properties as the PET fabric. Therefore, the grafting of hydroxyl can improve the hydrophobic effect of LMA coating and the binding property between LMA and fabrics effectively, without changing the wearing comfort.

**Keywords:** superhydrophobic; polyethylene glycol (PEG); hydroxyl; stable; lauryl methacrylate (LMA)

## 1. Introduction

In 1997, Barthlott and Neinhuis [1] discovered the unique self-cleaning properties of lotus leaf, and electronic microscopy of the surface of lotus leaves showed protruding nubs of about 20–40 μm, covered with smaller-scale roughness [2]. A lot of studies confirmed that the combination of nano- or micro-roughness, along with low surface energy, could result in a water contact angle higher than 150° [3]. Surfaces with these properties are called "superhydrophobic". Cotton and polyester fabric are the most widely-used fabrics in our daily lives and industries. Thus, an enormous amount of researchers are interested in incorporating the superhydrophobic cotton or poly(ethylene terephthalate)

(PET) fabrics [4] into a vast number of applications, such as clothing, moisture collection [5] and corrosion-resistance [6], etc.

With years of efforts, researchers found ways to create superhydrophobic surfaces by tailoring the surface topography and chemical composition by using various techniques, such as layer-by-layer assembling [7], electrochemical deposition treatment [8], the sol-gel method [9], electrospinning [10], dip-coating [11], and their combinations [12]. Nevertheless, the aforementioned methods usually rely on multi-step and time-consuming processes necessary for nanoparticle synthesis or functionalization. Furthermore, they share the common disadvantages of involving many reactive chemicals, especially volatile organic compound (VOC) that may pose health risks. All these techniques to make superhydrophobic surfaces can be simply divided into two functions: to make a rough surface and to lower surface energy.

Recently, as one of the environmentally friendly processes, plasma treatment has become more and more popular in modifying the surface properties of polymers and textile materials. The reaction occurs in non-equilibrium plasma with high electron density and high power density, and the vaporized monomer substances are expected to suffer from molecular dissociation, excitation, and ionization, caused by intense electron collision [13]. It is possible that the electron impact results in monomer chain scission, producing smaller fragmented radicals. In addition, the collision would create radicals in the polymer side chain due to the methyl abstraction process. This permits the formation of branching or networked structures [14]. Another competing chemical process taking place simultaneously is plasma polymerization, which combines those dissociated radicals and forms a disordered 3-D structure on the surface. Since the reactions take place at relatively high energy and in high-entropy states, the subsequent structures are highly irregular with a high degree of roughness, favorable to superhydrophobic properties [15]. Compared with traditional methods, plasma treatment has many advantages [16]: (1) it only modifies the thinnest layer of the surface, without changing the bulk properties of the materials; (2) it has lower chemical consumption and higher security; (3) it is totally fit to the definition of ecological textile manufacturing, because it causes no production of waste water, which produces less of a burden on the environment [17].

Based on the above advantages, plasma treatment is totally fit to the definition of ecological textile manufacturing and can be used to manufacture superhydrophobic fabrics. Karaman et al. [18] presented the plasma polymerization of poly(hexafluorobutyl acrylate) (PHFBA) thin films on different substrates in an RF (radio-frequency discharge) plasma reactor with an outer planar electrode, and they observed that a better hydrophobic property was obtained at high plasma power with a water contact angle of 156°, and they also confirmed that the monomer had covalent bond linkages to the fabric surface. Park et al. [19] fabricated a superhydrophobic PET fabric through the etching and deposition of hexamethyldisiloxane (HMDSO) via the oxygen plasma-enhanced chemical vapor deposition (PECVD) method. After PECVD treatment, the surface of the PET fabric obtained multi-scale roughness, and the contact angle with water and isopropanol solution was higher than 160°. The water contact angles of the PET the fabrics, which unetched or plasma-etched for one minute, were 144° and 160°, respectively. However, with a prolonged plasma-etching time (>1 min), the aspect ratio of the nano-convex on the surface of the PET fabric would become too large, thus causing the reduction in the water repellency of the PET fabric. Based on this observation, they pointed out that the fabric itself contains a micro-scale structure due to the micro-scale diameter of yarns, and the nano-scale roughness played a leading role in the superhydrophobicity of the fabric. Liu et al. [20] prepared superhydrophobic cotton fabric with octadecyltriethoxysilane (ODTMS) by argon plasma treatment. The water contact angle and sliding angle of ODTMS-coated cotton fabrics with or without plasma treatment were 149°, 9°, 154° and 4.5°, respectively. In addition, the ODTMS -coated cotton fabrics without plasma treatment had poor washing durability and lost their superhydrophobic properties after 10 washing cycles, whereas after plasma treatment they still exhibited superhydrophobicity after 150 washing cycles. Analysis found that the formation of Si–O–Si bonds on the PET fabric's surface after plasma treatment enhanced the water repellency and washing durability of cotton fabrics.

In our past research, we used dodecyl methacrylate (LMA) as the monomer via the plasma-enhanced chemical vapor deposition (PECVD) method [21–23] and the immersion-plasma-induced crosslinking method [24], respectively, to successfully prepare durable superhydrophobic cotton fabric, and it was found that the superhydrophobicity of cotton fabric made by the immersion-plasma-induced crosslinking method had better durability. Therefore, in this paper, cotton, PET, thermoplastic polyurethane (TPU) and polyamides-6 (PA6) fabrics were treated by LMA and low-voltage capacitively coupled discharge plasma (LP-CCP) via the immersion-plasma-induced crosslinking method, and the performances of the cotton and PET fabrics were compared to probe the influence of the presence of hydroxyl groups on the properties of LMA-coated fabrics.

## 2. Experimental Procedure

The knitted polyester fabrics (100%, 120 g/m$^2$), double-knitted cotton fabrics (100%, 210.85 g/m$^2$), plain woven thermoplastic polyurethane fabrics (100%, 119 g/m$^2$) and plain-woven polyamides-6 fabrics (100%, 196 g/m$^2$) were purchased from Miandu Textile Co., Ltd. (Nantong, China), and were used as samples. Polyethylene glycol (PEG-1000, CAS# 25322-68-3) and lauryl methacrylate (LMA, 97%, CAS# 142-09-6) were supplied by Alfa Aesar Tech. Co., Ltd. (Shanghai, China) as the monomer. Detergent 209 and standard soapflake were purchased from Wangnilai Co., Ltd. (Guangzhou, China) and the China Textile Institute of Science and Technology (Beijing, China) respectively. Ethanol (AR, ≥99.7%) and argon gas were supplied by Changzhou Hongsheng Fine Detail Co., Ltd. (Changzhou, China) and Canghai industry Gas Co., Ltd. (Shanghai, China), respectively.

All fabrics were washed to remove any possible dust or chemical residues, which can probably affect the surface treatment. The PET, TPU and PA6 fabrics were immersed in 2 g/L detergent 209 and 2 g/L sodium carbonate with the liquor ratio of 50:1 at a temperature of 40 °C in an ultrasonic bath for 40 min, and then washed repeatedly with deionized water and dried in an oven at a temperature of 70 °C for 2 h. The preprocessing method for cotton fabrics and the solution impregnation and plasma treatment for superhydrophobic fabrics have been described in the previous article [24].

Surface morphology was characterized by scanning electron microscope (SEM, Hitachi S-4800, Hitachi, Japan) and atomic force microscope (AFM, 5500AFM-SPM, Aglient, Santa Clara, CA, USA). The surface chemical compositions of samples were analyzed by X-ray photoelectron spectroscopy (XPS, Kratos AXIS UltraDLD, Shimazu, Japan) and attenuated total reflectance flourier transform infrared spectrometer (ATR-FTIR, Nicolet 6700, Waltham, MA, USA). The water repellency of the control and treated PET fabrics was evaluated by static water contact angle at room temperature and ambient humidity on the DropMeter™ Professional A-200 instrument (Ningbo Haishu Mai Detection Technology co. LTD, Ningbo, China), equipped with a video camera; the volume of droplets was 5 µL. The washing and abrasion durability of samples was evaluated by the water contact angles after several washing (according to AATCC 61-2006 [25]. 2A, SW-8, Mebon Instrument Co., Ltd., Chang Zhou, China) or abrasion (according to ISO 105-X12:2001 [26], dry friction mode, Y571N, Nantong Hongda Instrument co., Nantong, China) cycles, respectively. The wearing comfort of the fabric was determined by the water vapor transmission (GB/T 12704.2-2009 [27], YG601H, Ningbo Textile Instrument Factory, Zhejiang, China) and air permeability (GB/T5453-1997 [27], YG461E, Wenzhou Fangyuan Instrument Co., Ltd., Zhejiang, China). The mechanical strength of the samples was determined on an YG026MB multi-function electronic fabric strength machine (according to GB/T3923.1-2013 [28]). The moisture regains and wicking property were used to measure the hydrophilicity of fabrics according to GB/T 6503-2017 [29] and FZ/T 01071-2008 [30], respectively, and all the samples were conditioned for 48 h in atmospheric conditions of 20 ± 2 °C temperature and 65% ± 2% relative humidity before tests were performed.

## 3. Results and Discussion

### 3.1. Properties of LMA Coated Fabrics

In order to ensure the optimal monomer concentration of LMA, the fabrics were treated by plasma after being immersed in the monomer solution at different concentrations. The relationship between the surface water repellency of fabrics and the monomer concentration was as shown in Figure 1a. After treatment, the fabrics exhibited hydrophobicity with all monomer concentrations, indicating that the use of LMA can effectively improve the hydrophobicity of cotton and PET fabrics. Cotton and PET fabrics showed the best water repellency values after plasma treatment at LMA monomer concentrations of 5 and 35 g/L, respectively (the water contact angles of the cotton-*g*-LMA and PET-*g*-LMA fabrics were 156.7° and 152.6°, respectively). At the same time, washing and rubbing stability was also discussed, and the result is as shown in Figure 1b,c. It was found that the water contact angle of PET-*g*-LMA fabrics decreased rapidly (after 45 washing or 1000 rubbing cycles, the water contact angles reduce to 88.5°and 136.3°, respectively). However, Cotton-*g*-LMA fabrics exhibited better durability, and their water contact angles decreased slowly and stayed at 145.4 and 146.3°, after 45 washing or 1000 rubbing cycles, respectively. It was indicated that only using the proper amount of monomer, fabrics could get the best water repellency after plasma treatment. Moreover, the water repellency, and the washing and rubbing stability, of the cotton-*g*-LMA fabric were much better than those of the PET-*g*-LMA fabric. Therefore, in order to study the formation of this phenomenon, the PET fabrics were grafted with hydroxyl firstly and then coated with LMA, and the hydrophobicity and stability of these different PET fabrics were compared.

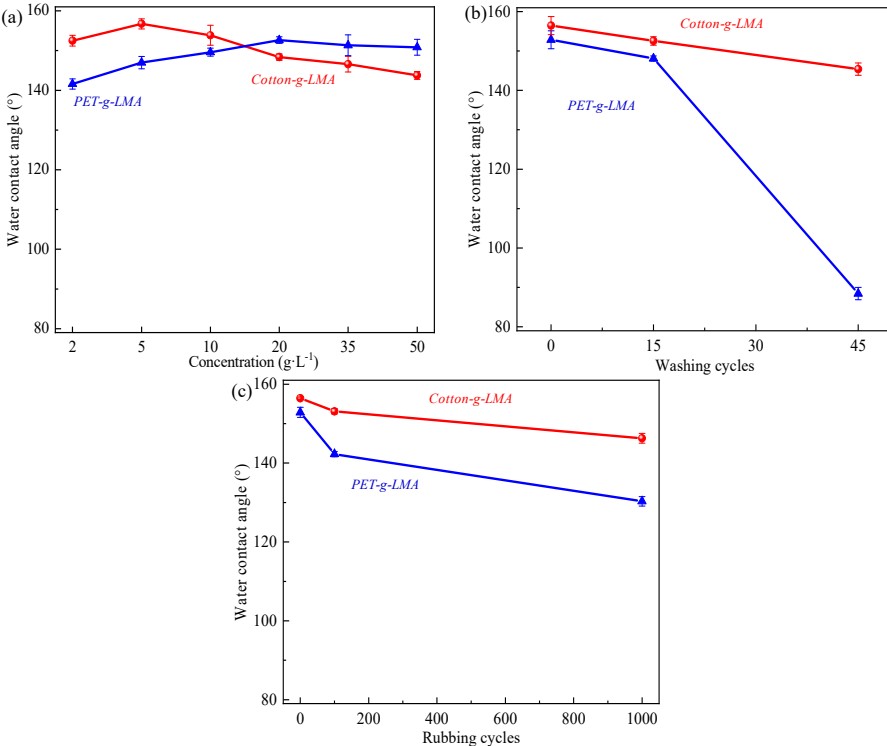

**Figure 1.** (**a**) Water repellency, (**b**) washing durability and (**c**) rubbing stability of Cotton-*g*-LMA and PET-*g*-LMA fabrics.

### 3.2. Hydrophilicity and Chemical Compositions

As shown in Figure 2a, after being PEG-coated, the moisture regains and wicking performance of PET fabrics increased from 0.40% and 0 mm/30 min to 0.91% and 95.30 mm/30 min, respectively. What is more, even after 20 washing cycles, the moisture regains and wicking performance of PET-*g*-PEG

fabrics still remained at 0.86% and 73.50 mm/30 min, which was much higher than PET fabrics and even just reduced slightly compared to PET-*g*-PEG fabrics. In addition, compared with cotton fabrics, PET-*g*-PEG fabrics have lower moisture regains but higher wicking performance, which indicates that after plasma treatment PET fabrics develop an excellent water repellency. According to the surface morphology of PET fabrics, PET-*g*-PEG fabrics and PET-*g*-PEG fabrics, after 20 washing cycles (shown in Figure 2b), the PEG film remained evenly and completely covered on the PET fibers' surface even after 20 washing cycles. Moreover, the integrity of the PEG film on the surface of the PET fibers was the main structural factor that ensured the hydrophilicity of the PET-*g*-PEG fabric. To determine the effects of different treatments on the chemical composition of PET fabrics, ATR-FTIR was used, and the result was as shown in Figure 2c. It was found that, after PEG impregnation and plasma crosslinking, the stretching vibration peak of the alcohol hydroxyl group was in the hydrogen bond association state, at about 3471 cm$^{-1}$ in region I, and the symmetric and antisymmetric stretching vibration peaks of –CH$_2$– at 3000~2800 cm$^{-1}$ in region II were all enhanced significantly, and were the characteristic peaks of PEG-1000 [31]. These results indicated that after plasma treatment, PET fabrics were successfully coated with PEG film, and developed excellent hydrophilicity and durability.

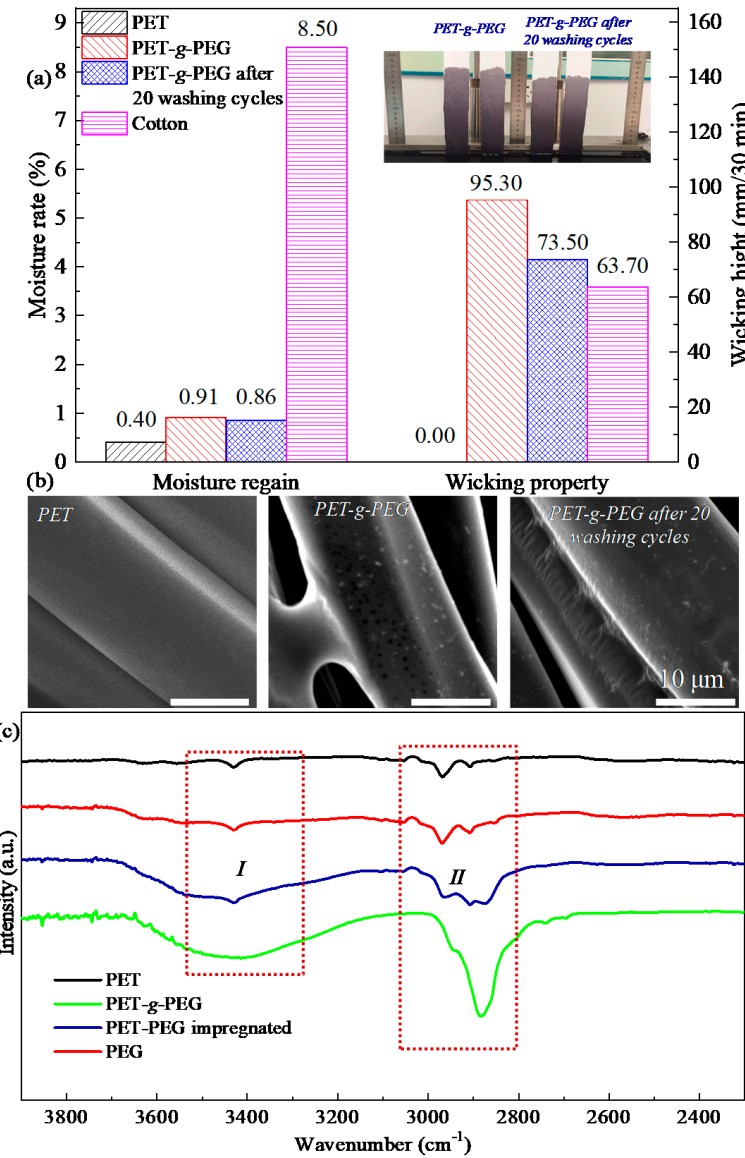

**Figure 2.** (**a**) The hydrophilicity, (**b**) surface morphology and (**c**) FTIR spectra of different fabrics.

### 3.3. Water Repellency of Fabrics

The water repellency of fabrics was generally determined by the contact state with water droplets, the water contact angle and the dipping state of the water. As shown in Figure 3, water has different contact states with different PET fabrics. When the PET fabric was in contact with water, it was in the Wenzel state; the water contact angle was about 121° (Figure 3a) and was suspended in water (Figure 3d), which was caused by the inherent hydrophobicity of the PET fabric. For cotton and PET fabrics after PEG impregnation or coating treatment, the fabrics were penetrated immediately by water with higher hydrophobicity (water contact angle was lower than 90°) (Figure 3b) and were completely immersed in water, even becoming soaked and sinking to the bottom of the test tube (Figure 3d). However, on the PET-*g*-LMA, PET-*g*-PEG & LMA and cotton-*g*-LMA fabric surfaces, the micro/nano composite rough structure can capture a stable air cushion to form a uniform and stable air shield on the surface. This makes it difficult for water molecules to get close to and adhere to the fabrics surface. In addition, when the PET-*g*-LMA, PET-*g*-PEG & LMA and cotton-*g*-LMA fabrics were in contact with water, they exhibited the Cassie state, with water contact angles higher than 150° (Figure 3c) [24]. The existence of the air shield on the superhydrophobic structure also made the fabric hang on the water surface (Figure 3d), and for cotton fabrics this has been discussed in our previous work. The cotton-*g*-LMA fabric floated on the surface [32]. This suggests that the PET-*g*-LMA, PET-*g*-PEG & LMA and cotton-*g*-LMA fabrics all have strong water repellency.

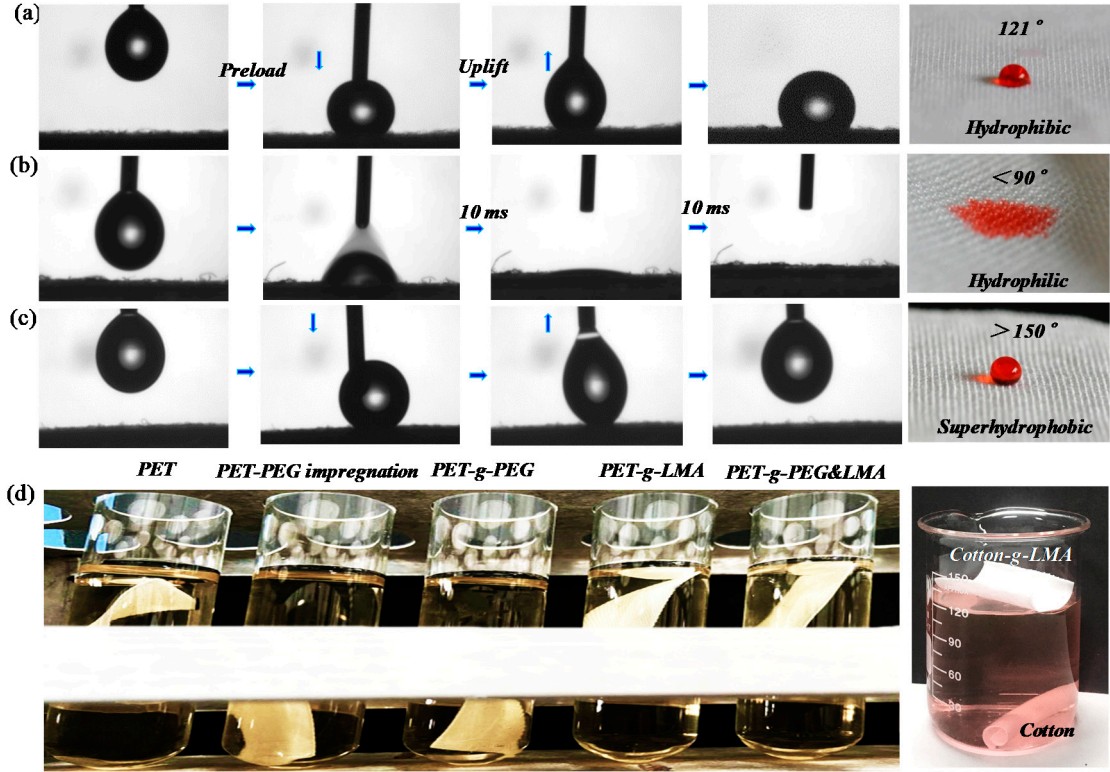

**Figure 3.** The contact states of water droplets with different PET fabric surfaces: (**a**) PET, (**b**) PET-g-PEG (or cotton), (**c**) PET-g-PEG&LMA (or PET-g-LMA or cotton-*g*-LMA) and (**d**) the state of different fabrics in water.

### 3.4. Mechanical Stability

The mechanical stability of superhydrophobic fabrics plays a vital role in their applications, such as the surface micro- or nano-scale roughness, which is mechanically weak and readily abraded [33]. So, in this paper, the durability under washing and rubbing were evaluated. Figure 4 shows the water repellency (Figure 4a,b) and surface morphology (Figure 4c–f) of different fabrics after different

washing or rubbing cycles, and it can be seen from Figure 4a,b that PET-*g*-PEG & LMA fabrics had almost the same washing and rubbing stability as cotton-*g*-LMA fabrics (after 45 washing cycles or 1000 rubbing cycles, the water contact angle remained at 145°, 134° and 146°, 143°, respectively), which was much higher than that of PET-*g*-LMA fabrics (after suffering 45 washing cycles or 1000 rubbing cycles, the water contact angle reduced rapidly, and the corresponding water contact angles were just 88° and 127°). Combined with SEM analysis (Figure 4b–f and Figure S1), all the superhydrophobicity properties of plasma-treated PET fabrics could be due to the drafting of the polymerized LMA film with chemical bonds under the plasma process. Moreover, the uniform structure of the film can help it maintain good hydrophobicity after 1500 rubbing cycles or 100 washing cycles. In addition, this paper also performed the same treatment on TPU and PA6 fabrics, which further verified that the introduction of hydroxyl groups could effectively increase the hydrophobicity and mechanical stability of the immersion-plasma-induced crosslinked and LMA-coated fabrics (Table S1).

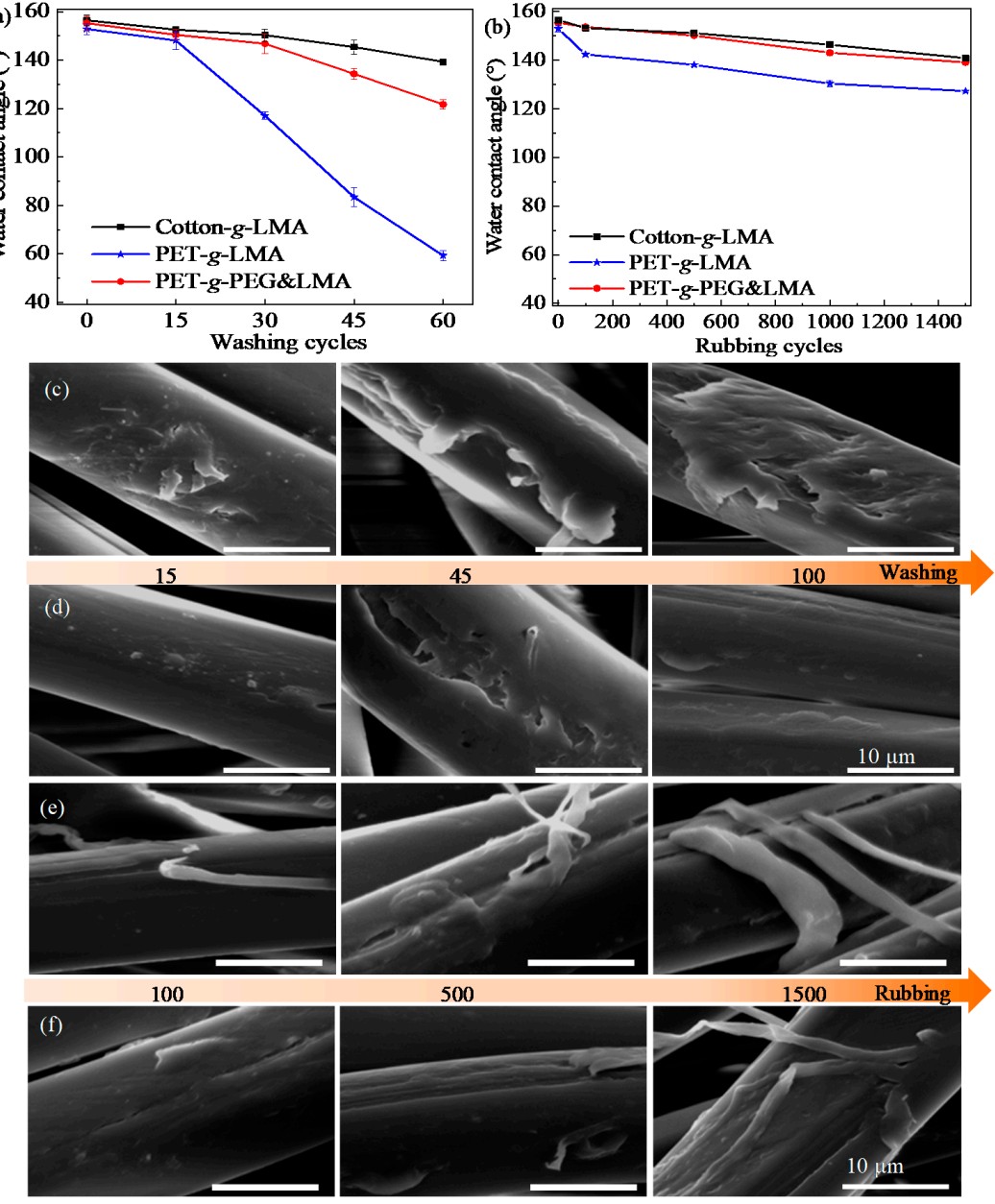

**Figure 4.** (**a**) The washing and (**b**) rubbing durability of PET fabrics and the corresponding surface morphologies of (**c**,**e**) PET-*g*-LMA and (**d**,**f**) PET-*g*-PEG & LMA fabrics.

### 3.5. Tensile Property

Tensile strength and breaking elongation were used to measure the tensile properties of PET fabrics, as exhibited in Table 1. As we can see, the PET fabric sample coated with LMA and PEG & LMA had a higher tensile strength and breaking elongation than the controlled one, which increased 20.55%, 21.12%, 13.34% and 20.52%, respectively. From the surface morphology in Figure S1, the increase in tensile strength might be due to the increase in surface roughness, caused by plasma etching and the deposition of monomer. Malkov [34] and Shen [28] pointed out that the RF plasma treatment had only a mild destructive effect on the surface of the fabric, and also, the etching treatment of plasma increased the friction between fibers, thereby ultimately creating stronger fabrics. Besides, the formation of uniform film on the surface of the PET fabrics increased the adhesion between the fibers, making it more difficult for the yarn to slide under the action of external forces. Therefore, the tensile property of the coated PET fabrics was improved to a certain extent. Vahid [35] derived a similar conclusion to us, finding that the plasma treatment could increase the tensile strength of fabrics.

**Table 1.** The tensile properties of different PET fabrics.

| Sample | Tensile Strength (N) | Breaking Elongation (%) |
|---|---|---|
| PET | 513.67 | 103.71 |
| PET-*g*-LMA | 619.21 | 117.54 |
| PET-*g*-PEG & LMA | 622.17 | 124.99 |

### 3.6. Wearing Comfort

The pore size distribution and porosity of the fabric are the main factors affecting the air permeability and water vapor transmission of fabrics, which are closely related to the wearing comfort. Therefore, we tested the air permeability and water vapor transmission of different PET fabrics. As shown in Table 2, compared to PET fabrics, the air permeability and water vapor transmission of PET-*g*-LMA and PET-*g*-PEG & LMA fabrics were reduced by 18.93%, 16.06%, 7.79% and 11.79%, respectively. This was due to the reduction in pore size and porosity of PET fabrics after the coating treatment, and the adhesion of the hydrophobic LMA film affected the absorption of water vapor by the PET fibers and the transportation to the low vapor pressure side of through capillary action between pores. What is more, PET-*g*-PEG & LMA and PET-*g*-LMA fabrics had extremely similar wearing comfort, which means that the grafting of hydroxyl groups onto the surfaces of PET fabrics could effectively improve their water repellency and mechanical durability without changing the wearing comfort and tensile property of the PET fabrics.

**Table 2.** The air permeability and water vapor transmission of different PET fabrics.

| Sample | Air Permeability (mm/s) | Water Vapor Transmission ($g\ m^{-2}\ h^{-1}$) |
|---|---|---|
| PET | 1026.61 | 48.41 |
| PET-*g*-LMA | 832.26 | 44.64 |
| PET-*g*-PEG & LMA | 861.69 | 42.70 |

## 4. Conclusions

In this paper, a superhydrophobic fabric was produced via the immersion-plasma crosslinking method, in the presence of the dodecyl methacrylate (LMA) monomer. In addition, we also discussed the influence of the existence of hydroxyl on the water repellency and mechanical stability of LMA-coated fabrics. It was found that the use of LMA could effectively improve the hydrophobicity of cotton and PET fabrics, with cotton-*g*-LMA fabrics exhibiting better durability and whose water contact angle decreased slowly and remained at 145.4° and 146.3°, after 45 washing or 1000 rubbing cycles, respectively. It was also shown that, after coating with PEG-1000, the hydroxyl was grafted onto the

surface of the PET fabrics successfully; in addition, the moisture regains and wicking performances of PET fabrics increased, and were well maintained even after 20 washing cycles. Moreover, PET-*g*-PEG & LMA fabrics had almost the same water repellency, and washing and rubbing stability, as cotton-*g*-LMA fabrics, which were much greater than those of PET-g-LMA fabrics. The water contact angles of Cotton-g-LMA, PET-g-LMA and PET-g-PEG & LMA were 156°, 153° and 155°, respectively, and after 45 washing cycles or 1000 rubbing cycles, the corresponding water contact angles decreased to 145°, 88° and 134°, and 146°, 127° and 143°, respectively. The uniform structure of the film can help it maintain good hydrophobicity after 1500 rubbing cycles or 100 washing cycles. Additionally, the PET-*g*-PEG & LMA and PET-*g*-LMA fabrics had extremely similar tensile properties and wearing comfort, proving that the grafting of hydroxyl groups onto the surface of the PET fabrics could effectively improve their water repellency and mechanical durability without changing their tensile property and wearing comfort. In addition, this paper also performed the same treatment with thermoplastic polyurethane (TPU) and polyamides-6 (PA6) fabrics, which further verified that the introduction of hydroxyl groups could effectively increase the hydrophobicity and mechanical stability of the immersion-plasma-induced crosslinked and LMA-coated fabrics.

**Supplementary Materials:** The following are available online at http://www.mdpi.com/2079-6412/10/12/1263/s1, Figure S1: The surface morphology of treated PET fabrics: (a) dipping PEG; (b) PET-g-PEG; (c) PET-g-LMA; (d) PET-g-PEG & LMA, Table S1: Water repellency and washing stability of different fabrics.

**Author Contributions:** Y.G., R.Z. and Y.S. contributed to the project management, L.X. and Y.Z. contributed to the experiment and characterization of fabrics; L.X., J.S. and J.Y. contributed to the choice of materials. All authors have read and agreed to the published version of the manuscript.

**Funding:** This research was funded by the Natural Science Foundation of China (51702167), The Natural Science Foundation of the Jiangsu Higher Education Institutions of China (17KJA540001), Natural Science Foundation of China (Grant Nos. 11875104 and 11475043), National Key R&D Program of China (Project No.2017YFB0309100).

**Acknowledgments:** The authors would like to thank Chengjiao Zhang of School of Textile and Clothing, Nantong University for his valuable comments and guidance throughout this study.

**Conflicts of Interest:** The authors declare no conflict of interest.

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
