# Peer review of "The Effect of Hydroxyl on the Superhydrophobicity of Dodecyl Methacrylate (LMA) Coated Fabrics through Simple Dipping-Plasma Crosslinked Method"

_coatings, doi:10.3390/coatings10121263_

Round 1
Reviewer 1 Report
The paper is interesting, but it has to be revised and/or upgraded. After that, I suggest that the paper may be accepted for publication in Coatings.
The work, presented in this paper, presents comprehensive research on a particular topic.
1. In the Experimental part of the paper (Page 3, line 107-108) the authors mentioned that one of the testing methods would be investigation of the tensile properties of fabrics, according to GB/T3923.1-2013, i.e, Determination of maximum force and elongation at maximum force using the strip method. I cannot see those results in the paper, so it is necessarily to include them. Additionally, it has to be explained and connected with other results.
Thanks for your kind advice. We have added the tensile properties in paper (Page 6, line 213-226), and the detail was as follows:
3.5. Tensile property
Tensile strength and breaking elongation was used to measure the tensile property of PET fabrics, which was exhibited in Table 1. As we can see, the PET fabric sample coated by LMA and PEG & LMA had higher tensile strength and breaking elongation than the controlled one, and it increased 20.55%, 21.12% and 13.34%, 20.52%, respectively. From the surface morphology in Figure S1, the increase of tensile might be due to the increase of surface roughness, caused by plasma etching and the deposition of monomer. Malkov [29] and Shen [30] had pointed that, the RF plasma treatment had only a mild destructive effect on the surface of the fabric, also, the etching treatment of plasma increased the friction between fibers and thereby, ultimately creates stronger fabrics. Besides, the formation of uniform film on the surface of PET fabrics increased the adhesion between the fibers makes it more difficult for the yarn to slide under the action of external forces. Therefore, the tensile property of the coated PET fabrics was improved to a certain extent. Vahid [31] got the similar conclusions to us that plasma treatment could increase the tensile strength of fabrics.
Table 1. The tensile property of different PET fabrics.
|
Sample |
Tensile strength (N) |
Breaking elongation (%) |
|
PET |
513.67 |
103.71 |
|
PET-g-LMA |
619.21 |
117.54 |
|
PET-g-PEG&LMA |
622.17 |
124.99 |
2. Page 5: Figure 3, I suggest enlarging the image a bit, so that the results are clearly visible
Thanks for your kind advice. We have enlarged the image as required.
Reviewer 2 Report
1. Section 3.2, 3.3 & 3.5 discussed only PET fabrics, why?
Thanks very much for the suggestion.
We mainly discussed the effect of the presence of hydroxyl on the super-hydrophobicity and durability of dodecyl methacrylate (LMA) coated fabrics. Therefore, in Section 3.2 and 3.3, the grafting of hydroxyl and hydrophilic treatment of PET fabric surface. To clarify the main points of 3.2 and 3.3, we added the following paragraph at the end of 3.1 in line 148-151:
Therefore, in order to study the formation of this phenomenon, PET fabrics was grafted hydroxyl firstly and then coated with LMA, and the hydrophobicity and stability of these different PET fabrics were compared.
Section was illustrated the impact of hydrophilic treatment on the wearing comfort of hydrophobic PET fabrics. Due to the difference in inherent properties between PET and cotton fabrics, the wearing comfort of cotton fabrics was not discussed in this section. And the corresponding data was as shown in follow:
The air permeability and water-vapor transmission of different fabrics.
|
Sample |
Air permeability (mm/s) |
Water-vapor transmission(g/(m2·h) |
|
PET |
1026.61 |
48.41 |
|
PET-g-LMA |
832.26 |
44.64 |
|
PET-g-PEG&LMA |
861.69 |
42.70 |
|
Cotton-g-LMA |
466.79 |
94.34 |
2. As the manuscript claimed to compare the performance of cotton vs. PET fabrics coated with LMA but lacks the comparison.
Thanks for your kind advice. This paper was focused on the effect of hydroxyl on the super-hydrophobicity of LMA coated fabrics, and we have discussed the water repellency and durability of the surface of LMA coated cotton anf PET fabrics in Section 3.2 and 3.4.
Reviewer 3 Report
The aim and original contribution of the reported study are clearly described. So, the paper is focusing on the producing of superhydrophobic fabrics via immersion-plasma crosslinking method, in the presence of the dodecyl methacrylate (LMA) monomer. It was found that the use of LMA can effectively improve the hydrophobicity of cotton and PET fabrics, with Cotton-g-LMA fabrics exhibiting better durability and whose water contact angle decreased slowly and kept at 145.4 and 146.3°, after 45 washing or 1000 rubbing cycles, respectively. I was also shown that, after coating with PEG-1000, the moisture regains and wicking performance of PET fabrics increased and were well maintained even after 20 washing cycles. Moreover, the uniform structure of the film can help it still maintain good hydrophobicity after 1500 rubbing cycles or 100 washing cycles. Also, PET-g-PEG&LMA and PET-g-LMA fabrics had extremely the same wearing comfort, proving that the grafting of hydroxyl groups on the surface of PET fabrics could effectively improve their water repellency and mechanical durability without changing their wearing comfort.
In addition, this paper also performed the same treatment of thermoplastic polyurethane (TPU) and polyamides-6 (PA6) fabrics respectively, which further verified that the introduction of hydroxyl groups could effectively increase the hydrophobicity and mechanical stability of the immersion-plasma induced crosslinking LMA coated fabrics.
However, I have a few small remarks:
1. The conclusion section should be improved.
Thanks very much for the suggestion. We have reviewed and written the conclusion again in Page 10-11 Line 243-264, and the detail was as follows:
In this paper, a superhydrophobic fabric was produced via immersion-plasma crosslinking method, in the presence of the dodecyl methacrylate (LMA) monomer. In addition, we also discussed the influence of the existence of hydroxyl to the water repellency and mechanical stability of LMA coated fabrics. It was found that the use of LMA could effectively improve the hydrophobicity of cotton and PET fabrics, with cotton-g-LMA fabrics exhibiting better durability and whose water contact angle decreased slowly and kept at 145.4 and 146.3°, after 45 washing or 1000 rubbing cycles, respectively. It was also shown that, after coating with PEG-1000, the hydroxyl was grafted on the surface of PET fabrics successfully, in addition, the moisture regains and wicking performance of PET fabrics increased and were well maintained even after 20 washing cycles. Moreover, PET-g-PEG&LMA fabrics had almost the same water repellency, washing and rubbing stability to that of cotton-g-LMA fabrics, which were much higher than that of PET-g-LMA fabrics. The water contact angle of Cotton-g-LMA, PET-g-LMA and PET-g-PEG&LMA was 156°, 153° and 155°, respectively, and after 45 washing cycles or 1000 rubbing cycles, the corresponding water contact angle was decreased to 145°, 88°, 134° and 146°, 127°, 143°, respectively. The uniform structure of the film can help it still maintain good hydrophobicity after 1500 rubbing cycles or 100 washing cycles. Also, PET-g-PEG&LMA and PET-g-LMA fabrics had extremely the same tensile property and wearing comfort, proving that the grafting of hydroxyl groups on the surface of PET fabrics could effectively improve their water repellency and mechanical durability without changing their tensile property and wearing comfort. In addition, this paper also performed the same treatment of thermoplastic polyurethane (TPU) and polyamides-6 (PA6) fabrics respectively, which further verified that the introduction of hydroxyl groups could effectively increase the hydrophobicity and mechanical stability of the immersion-plasma induced crosslinking LMA coated fabrics.
2. In the supplementary table (Table S1) there is a column entitled “After 500 rubing cycles”, without providing the values of the measured water contact angles. The author should either provide the values or remove the column.
Thanks for your kind advice. We have removed the column of “After 500 rubing cycles” in Table S1.
Round 2
Reviewer 1 Report
Thank you for accepting the suggestions, correcting and supplementing the paper. It is well-done now.
Dear Reviwer,
Thank you very much for your affirmation!
Best regards
Reviewer 2 Report
The manuscript got a significant revision. However, it still needs to address the following comments.
1. Though the authors explained how LMA modifying the properties of PET. But, the paper needs to include the effect of LMA on cotton as well.
Thanks very much for the suggestion. As the properties of cotton-g-LMA fabrics have been described in detail in our previous articles, therefore, we added some cotton fabric properties related to to this paper as suggestion in Sections 3.2 and 3.3, and the detailed instructions are given in the next point.
2. The effect of LMA on cotton should include in Sections 3.2 & 3.3 besides PET.
Thanks for your kind advice. We have modified Sections 3.2 and 3.3 as suggested in the paper, and the details were as follows:
As shown in Figure 2a, after PEG coated, the moisture regains and wicking performance of PET fabrics increased from 0.40% and 0 mm/30 min to 0.91% and 95.30 mm/30 min. What’s more, even after 20 washing cycles, the moisture regains and wicking performance of PET-g-PEG fabrics still retained at 0.86% and 73.50 mm/30 min, which was much higher than PET fabrics and even just reduced slightly to PET-g-PEG fabrics. In addition, compared with cotton fabrics, PET-g-PEG fabrics have lower moisture regains but higher wicking performance, which indicated that after plasma treated PET fabrics got an excellent water repellency. According to the surface morphology of PET fabrics, PET-g-PEG fabrics and PET-g-PEG fabrics after 20 washing cycles in Figure 2b, PEG film had been evenly and completely covered on the PET fibers surface even after 20 washing cycles. Moreover, the integrity of PEG film on the surface of PET fibers was the main structural factor to ensure the hydrophilicity of PET-g-PEG fabric. To determine the effect of different treatments on the chemical composition of PET fabrics, ATR-FTIR was used and the result was as shown in Figure 2c. It was found that, after PEG impregnated and plasma crosslinked, the stretching vibration peak of alcohol hydroxyl group in hydrogen bond association state at about 3471 cm-1 in region I, and the symmetric and antisymmetric stretching vibration peaks of -CH2- at 3000~2800 cm-1 in region II were all enhanced significantly, which were the characteristic peaks of PEG-1000 [25]. These results indicated that, after plasma treatment PET fabrics were successfully coated with PEG film and got excellent hydrophilicity and durability.
The water repellency of fabrics was generally determined by the contact state with water droplet, water contact angle and the dipping state with water. As shown in Figure 3, water has different contact states with different PET fabrics. When PET fabric was in contact with water, it emerged the Wenzel state, the water contact angle was about 121° (Figure 3a) and was suspended in water (Figure 3d), which was caused by the inherent hydrophobicity of PET fabric. For cotton and PET fabrics that after PEG impregnation or coating treatment, fabrics were penetrated immediately by water with higher hydrophobicity (water contact angle was lower than 10°) (Figure 3b) and completely immersed in water, even soaked and sunk to the bottom of the test tube (Figure 3d). However, in the PET-g-LMA, PET-g-PEG&LMA and cotton-g-LMA fabric surface the micro-nano composite rough structure can capture the stable air cushion to form a uniform and stable air shield on the surface, this makes it difficult for water molecules to get close to and adhere to the fabrics surface [26]. In addition, when PET-g-LMA, PET-g-PEG&LMA and cotton-g-LMA fabric was in contact with water, it emerged the Cassie state with water contact angle higher than 150° (Figure 3c) [27]. The existence of the air shield on the super-hydrophobic structure also made the fabric hanging on the deionized water surface (Figure 3d). And for cotton fabrics this has been discussed in our previous work and cotton-g-LMA fabric floated on the surface [25]. It suggested the PET-g-LMA, PET-g-PEG&LMA and cotton-g-LMA fabric all has strong water repellency.
Round 3
Reviewer 2 Report
Better